# Isofraxidin Attenuates Lipopolysaccharide-Induced Cytokine Release in Mice Lung and Liver Tissues via Inhibiting Inflammation and Oxidative Stress

**DOI:** 10.3390/biomedicines13030653

**Published:** 2025-03-07

**Authors:** Marwa Salih Al-Naimi, Ahmed R. Abu-Raghif, Ahmed F. Abed Mansoor, Hayder Adnan Fawzi

**Affiliations:** 1Department of Pharmacology, College of Medicine, Al-Nahrain University, Baghdad 10006, Iraq; ar_armat1967@nahrainuniv.edu.iq; 2Department of Pharmacology and Toxicology, College of Pharmacy, Al-Farahidi University, Baghdad 00965, Iraq; 3Department of Pharmacology and Toxicology, College of Pharmacy, National University of Science and Technology, Nasiriyah 64001, Iraq; ahmediraqiz1987_ah@yahoo.com; 4Department of Clinical Pharmacy, College of Pharmacy, AlMustafa University, Baghdad 10064, Iraq; hayder.adnan2010@gmail.com

**Keywords:** isofraxidin, cytokine storm, inflammation, antioxidant, steroid

## Abstract

**Background**: Isofraxidin is a hydroxylcoumarin derived from herbal Fraxinus and Eleutherococcus. It has been shown that isofraxidin has antioxidant, anti-inflammatory, anti-diabetic, and anti-lipidemic effects. The study aimed to examine the therapeutic effects of isofraxidin with and without methylprednisolone to ameliorate lipopolysaccharide (LPS)-induced cytokine-releasing syndrome. **Methods**: The study comprised two phases: preventive and therapeutic. In all the experiments that involved LPS induction, a single dose of LPS (5 mg/kg) was used. The preventive phase involved the administration of the agents before LPS induction, in which 50 mg/kg of methylprednisolone, 15 mg/kg of isofraxidin, or a combination of 7.5 mg/kg of isofraxidin plus 25 mg/kg methylprednisolone were given daily for 3 days before induction. The therapeutic phase involved the administration of the following agents after LPS induction: 50 mg/kg methylprednisolone, 15 mg/kg of isofraxidin, or a combination of 7.5 mg/kg of isofraxidin plus 25 mg/kg methylprednisolone were given once daily was given for 7 days. **Results**: Isofraxidin treatment with or without methylprednisolone ameliorates LPS-induced inflammatory and oxidative stress damage in mice; it reduces the inflammatory (IL-6, TNF-α, IL-1β, IL-8, Malondialdehyde, and IFN-γ) and oxidative stress markers. Additionally, isofraxidin treatment with or without methylprednisolone prevented liver and lung tissue damage induced by LPS. **Conclusions**: Isofraxidin exhibited preventive and therapeutic properties against lipopolysaccharide-induced cytokine storms in mice via anti-inflammatory and antioxidant pathways, and its combination with methylprednisolone demonstrated synergistic outcomes.

## 1. Introduction

Acute lung damage and acute respiratory distress syndrome are recognized as a category of pulmonary disease characterized by widespread inflammatory cell infiltration, hemorrhage, and pulmonary edema [1,2]. Factors that might directly or indirectly generate these lung disorders include inhalation, pneumonia, sepsis, trauma, pancreatitis, and transfusion, among others [3]. Numerous studies indicate that inflammation is pivotal to the etiology of acute lung injury. A substantial influx of immune cells congregates at injury sites, triggering a cascade of inflammatory signaling pathways and releasing several pro-inflammatory cytokines. Consequently, inflammation compromises the barrier function of the alveolar epithelium and vascular endothelium, thereby enhancing the permeability of the alveolar–capillary membrane. This results in the accumulation of protein-rich edema fluid in the lung interstitium, ultimately causing pulmonary edema and tissue damage [4,5].

Cytokine release syndrome (CRS), also known as cytokine storm, arises from the production of cytokines, especially in response to protein therapies that target immune cells. CRS engages the innate immune system, does not necessitate antibody formation, and can manifest upon initial exposure [6]; these agents induce CRS by prompting the rapid release of tumor necrosis factor (TNF)-α and interferon (IFN)-γ, succeeded by a subsequent release of interleukin (IL)-6, resulting in the extensive activation of several inflammatory mediators [7]. Increased levels of pro-inflammatory cytokines cause tissue and organ damage through systemic inflammatory processes, resulting in clinical presentations such as flu-like symptoms, fever, chills, pain, and, in severe instances, multiple organ failure and death [8]. Cytokine storms can also be induced following infection with specific bacteria under physiological settings [9].

IL-6 is recognized as the focal point of CRS and plays a pivotal role in inflammatory and immune responses. IL-6 conveys messages and exerts its effects through three principal signaling modalities: classic signaling patterns, trans-signaling pathways, and trans-presentation pathways [10,11]. The IL-6-mediated JAK/STAT3 pathway is essential for fully activating the NF-κB (nuclear factor kappa-light-chain-enhancer of activated B cells) pathway. Activating NF-κB can induce IL-6 production, leading to a cascading amplification effect [12]. In this mechanism, both IL-1 and TNF-α activate the NF-κB pathway, facilitating the production of IL-6 [13]. Consequently, IL-6 stimulates the synthesis of IL-1 and TNF-α by activating Th17 and other pathways in succession [14].

Lung damage is prevalent in CRS, particularly in severe respiratory infections. High IL-6 signaling can lead to the accumulation of innate and adaptive immune cells in the lungs [15]. Upon activation by cytokines such as TNF and IFN, these immune cells secrete substantial quantities of free radicals and proteases, resulting in an assault on capillary endothelial cells and lung epithelial cells [16].

Isofraxidin is a hydroxyl-coumarin derived from herbal *Fraxinus* and *Eleutherococcus* [17]. Isofraxidin has many pharmacological and biological effects, including neuroprotection, cardioprotection, anti-obesity, and anti-inflammatory effects. Isofraxidin affects different signaling pathways, including NF-κB, pro-inflammatory cytokines, cyclooxygenase 2 (COX-2), mitogen-activated protein kinase (MAPK), and inducible nitric oxide synthase (iNOS) [18]. Isofraxidin is rapidly absorbed after oral administration, reaches a plasma steady state within 20 h, and has a short half-life of about 4 h. The liver metabolizes isofraxidin to dihydroxyl-methoxy coumarin [18]. Isofraxidin is highly lipid-soluble and can cross the blood–brain barrier [18].

The anti-inflammatory effect of isofraxidin is related to its inhibiting of pro-inflammatory cytokines, including TNF-α, IL-1β, and IL-6. In addition, isofraxidin inhibits the expression of matrix metalloproteinases (MMPs) and disinters metalloproteinase and thrombospondin motif 4,5 (ADAMTS-4,5), augmenting pro-inflammatory cytokine expression. Furthermore, isofraxidin inhibits the expression of toll-like receptor 4 (TLR4), myeloid differentiation protein 2 (MD2), activating protein 1 (AP1), and extracellular signal-regulated protein kinases 1 and 2 (ERK1/2). These findings indicate that isofraxidin and related compounds have anti-inflammatory effects by inhibiting various inflammatory mediators [19]. Furthermore, it can scavenge reactive oxygen species (ROS) and prevent the development of oxidative stress. It has been shown that isofraxidin has antioxidant, anti-inflammatory, anti-diabetic, and anti-lipidemic effects by inhibiting the generation of ROS and increasing the expression of antioxidant enzymes. Being coumarin-derivative, isofraxidin reduces the development of oxidative stress via the modulation of oxidative stress and associated lipid peroxidation [20,21].

No study in the literature has examined the effect of isofraxidin on CRS. Due to its multiple mechanisms of action, including anti-inflammatory and antioxidant activity, these actions could benefit CRS, which develops due to an exaggerated immune response and is associated with hyper-inflammation and hypercytokinemia, targeting inflammatory signaling pathways and pro-inflammatory cytokines via the pleiotropic anti-inflammatory action of isofraxidin. The current work addressed this knowledge gap; the study examined the therapeutic effects of isofraxidin with or without methylprednisolone to ameliorate lipopolysaccharide (LPS)-induced cytokine-releasing syndrome.

## 2. Materials and Methods

### 2.1. Experimental Animals

Swiss albino mice (n = 100) were obtained from the Ministry of Health and Environment, Center for Drug Control and Research. The mice weighed 30–40 g and were 6–9 weeks old. All animal-related activities and experimentation processes were conducted strictly within the rules for the care and use of laboratory animals established by the animal ethics committee at Al-Nahrain University, College of Pharmacy; all animal experiments were carried out following AVMA guidelines 2020 [22]. The authors adhered to the ARRIVE 2.0 guidelines [23].

### 2.2. Study Design

The study comprised two phases: preventive and therapeutic. In all the experiments that involved induction by LPS (Sigma-Aldrich, Darmstadt, Germany, *Escherichia coli*, serotype 055: B5, lot 0000133605/99%), a single dose of LPS (5 mg/kg) was administered intraperitoneally; all agents, including LPS, were administered intraperitoneally [24,25,26]. The preventive phase involved the administration of the agents before LPS induction; each group contained 10 animals. In the **P-MP** group, 50 mg/kg of methylprednisolone (Hangzhou Hyper Chem. Limited, Hangzhou, China) was given daily for 3 days before induction. After induction, the animals were left for 2 days without intervention [24,27]. Similar procedures were performed for the **P-ISO** (15 mg/kg once daily of isofraxidin (Hangzhou Hyper Chem. Limited, Hangzhou, China) for 3 days [28]) and **P-COM** groups (a combination of 7.5 mg/kg of isofraxidin plus 25 mg/kg methylprednisolone for 3 days [27,28]) before LPS induction. Additionally, two groups, **P-C** (negative control) and **P-I** (LPS induction), did not receive any treatment and were followed up after 5 days, as seen in Figure 1.

The therapeutic phase involved the administration of the agents after LPS induction; each group contained 10 animals. In the **T-MP** group, after 1 h of LPS induction, 50 mg/kg methylprednisolone once daily was given for 7 days [29,30]. Similar procedures were performed for the **T-ISO** (15 mg/kg once daily of isofraxidin for 7 days [28]) and **T-COM** groups (a combination of 7.5 mg/kg isofraxidin plus 25 mg/kg methylprednisolone for 7 days [28,29]) after LPS induction. Additionally, the **T-C** (negative control) and **T-I** (LPS induction) groups did not receive any treatment and were followed up after 7 days, as seen in Figure 1. The mice were anesthetized intraperitoneally with 80 mg/kg of ketamine and 10 mg/kg of xylazine, followed by the mice being euthanized by exsanguination through cardiac puncture [31,32,33]; blood and tissue samples were collected for further analysis.

### 2.3. Clinical Observations and Animal Care

Every attempt was undertaken to reduce the pain and the quantity of animals engaged in the investigations. The mice were observed both post-injection and 30 min later. In the event of bleeding, gauze was applied, and pressure was exerted. After the hemorrhage ceased, the area was sanitized with gauze and water. In instances of other complications, a veterinarian was consulted to evaluate the animal’s suitability for further participation in the experiment [34,35].

### 2.4. Laboratory Analysis of Mice Serum

All blood samples were obtained from the jugular vein, and plasma was separated using a centrifuge at 3000 rpm for 10 min; serum was collected in 2 mL Eppendorf tubes and was deposited at −20 °C for subsequent thawing [36]. The quantitative assessment of biomarkers (GSH, IL-1β, IL-8, IFN-γ, IL-6, TNF-α, and MDA) in mouse serum was conducted using enzyme-linked immunosorbent assay (ELISA) kits, following the manufacturer’s instructions (Sunlong, Shanghai, China).

The kits employ a sandwich ELISA method, incorporating two antibodies that target different antigen epitopes. Each plate was pre-coated with capture antibodies and subsequently filled with serum samples to facilitate a specific antigen–antibody interaction through incubation with an enzyme-conjugated antibody. Subsequently, a washout procedure was conducted to remove the unbound antibodies. Ultimately, the substrate was introduced to produce a calorimetric signal detectable by the plate reader (Diagnostic Automation, Cortez Diagnostics^®^, Los Angeles, CA, USA) [37].

### 2.5. Histopathology of Lung and Liver Tissue

The liver and lung organs were removed and processed via the formalin-fixed paraffin-embedded technique for the histological examination of alterations following cytokine storm induction and therapies. The liver and lung tissues were preserved in 10% neutral formalin for one night. The tissues were dehydrated in ethanol, cleaned with xylene, and subsequently embedded in paraffin blocks using conventional protocols. Five-micrometer-thick paraffin sections were prepared using a microtome. Sections were stained with hematoxylin and eosin (H&E) [38]. Two independent pathologists, blind to the study groups, evaluated the grading of disease activity and assigned numerical values to this semiquantitative score.

The entire liver architecture was evaluated at 10× and 40× magnification; the damage score was based on four characteristics: (1) signified congestion, (2) signified edema, (3) signified infiltration by polymorphonuclear leukocytes, and (4) signified necrosis. The combined value of these scores was computed and designated as the total score at 40× magnification in 10 chosen regions of the prepared slide [39].

### 2.6. Sample Size Calculation and Animal Randomization

A post hoc sample size was decided upon with an effect size of 0.5 and an alpha level of 0.05; F-family tests with a total sample size of 40 for each group of 10 animals and the software program G.Power version 3.1 was employed to calculate the sample size [40,41].

### 2.7. Ethical Consideration

The study was approved by the Research Ethical Committee of the College of Medicine, Al-Nahrain University, with approval number (UNCOMIRB36902024) and data (4 December 2022).

### 2.8. Statistical Analysis

Descriptive and inferential statistical analyses were conducted with GraphPad Prism 10.3. The normality test (Anderson–Darling test) was initially used to assess whether the continuous variables followed a normal distribution. Variables were analyzed using ANOVA and post hoc Tukey tests. A significance level of 0.05 was applied throughout the study.

## 3. Results

### 3.1. Prevention of CRS by Isofraxidin With and Without Methylprednisolone

The serum levels of IL-6, TNF-α, IL-1β, IL-8, MDA, and IFN-γ were significantly higher in the induction group compared to the control group, highlighting the severity of CRS. Conversely, GSH levels were notably lower in the induction group. The data presented in Figure 2 are essential for understanding the preventive effects of the medications in the CRS context.

In the methylprednisolone, isofraxidin, and combination groups, serum concentrations of IL-6, TNF-α, IL-1β, IL-8, MDA, and IFN-γ were significantly reduced, while GSH levels were markedly increased compared to the induction group. Isofraxidin alone resulted in substantially elevated TNF-α, IL-6, IL-1β, IFN-γ, and MDA levels. However, GSH levels were significantly lower than in the P-MP group, except for IL-8, which did not show significant differences. As depicted in Figure 2, the combination of isofraxidin and methylprednisolone led to significantly reduced levels of IL-6, TNF-α, IL-1β, IL-8, MDA, and IFN-γ compared to the P-MP and P-ISO groups, with GSH levels being significantly higher.

### 3.2. Therapeutic Effects of Isofraxidin With and Without Methylprednisolone on CRS Inflammatory and Oxidative Stress Markers

The serum levels of IL-6, TNF-α, IL-1β, IL-8, MDA, and IFN-γ were markedly increased. GSH levels were markedly reduced in the induction group compared to the control group, as illustrated in Figure 3; this suggests the intensity of the cytokine storm triggered by LPS. Isofraxidin alone exhibited markedly elevated levels of IL-6, TNF-α, IL-1β, IL-8, MDA, and IFN-γ compared to the T-MP group. Isofraxidin, in conjunction with methylprednisolone, exhibited markedly reduced levels of IL-6, TNF-α, IL-1β, IL-8, MDA, and IFN-γ compared to the T-MP group, as illustrated in Figure 3. Isofraxidin exhibited markedly reduced GSH levels compared to the T-MP group. However, the combination of isofraxidin and methylprednisolone resulted in significantly elevated GSH levels relative to the T-MP group, as illustrated in Figure 3.

### 3.3. Lung Histopathology

The lung sections of un-treated mice displayed typical lung structures, featuring thin interalveolar septa, clear alveolar sacs, and regular alveolar septa with uniform air spaces. In the induction group, acute inflammation was pronounced and characterized by vascular congestion, capillary damage, thickened alveolar walls, constricted air spaces, and hyaline membrane formation. Mice given methylprednisolone before LPS induction showed modest interstitial inflammatory cell infiltration, mild vascular congestion, and intact alveolar spaces. For isofraxidin as a protective agent, mice in the isofraxidin + LPS group exhibited multifocal minor inflammatory cell infiltration, minimal vascular congestion and dilation, and some slight alveolar damage. The combination group also showed modest inflammatory cell infiltration, minimal vascular congestion and dilation, and a few instances of mild alveolar damage, as depicted in Figure 4.

Mice that underwent LPS induction and were subsequently treated with methylprednisolone showed mild to moderate interstitial inflammatory cell infiltration, slight vascular congestion, and intact alveolar spaces; regarding the therapeutic use of isofraxidin, mice in the isofraxidin group exhibited minor multifocal interstitial inflammatory cell infiltration, minimal vascular congestion and dilation, and minor alveolar damage. In contrast, mice in the combination group demonstrated mild inflammatory cell infiltration, slight vascular congestion and dilation, and preserved alveolar structures, as depicted in Figure 4.

### 3.4. Liver Histopathology

The standard liver sections stained with H&E showed portal regions containing the hepatic triad’s components: small branches of the portal vein, a branch of the hepatic artery, a small bile duct, lymphatic vessels, and minimal connective tissue. Liver cells were arranged in plates or cords originating from central venules. LPS induction caused significant vascular congestion and dilation, edema, multifocal mild mixed inflammatory cell infiltrations, and multifocal hepatocyte degeneration with necrosis. The mice treated with methylprednisolone before LPS induction exhibited vascular congestion, dilation, and mild mixed inflammatory cell infiltration. Regarding isofraxidin as a protective agent, mice in the isofraxidin + LPS group showed vascular congestion and dilation with edema, moderate mixed inflammatory cell infiltration, and moderate lobular hepatocyte degeneration with necrosis. In the isofraxidin + methylprednisolone and LPS group, mice displayed mild vascular congestion and dilation with edema, minimal inflammatory cell infiltration, and mild hepatocyte degeneration with minor necrosis, as shown in Figure 5A.

All groups significantly reduced overall liver scores compared to the induction group but still had significantly higher liver scores than the control group. The group treated with isofraxidin alone had markedly higher liver scores compared to the P-MP group. However, the combination of isofraxidin and MP showed no significant difference compared to the P-MP group, as shown in Figure 5B.

Regarding isofraxidin as a therapeutic agent, the mice treated with the LPS + isofraxidin combination showed vascular congestion and dilation, minor edema, limited mixed inflammatory cell infiltration, and moderate lobular hepatocyte degeneration with mild necrosis. The mice in the LPS + isofraxidin + MP group displayed vascular congestion and dilation with minor edema, slight inflammatory cell infiltration, and early hepatocyte degradation without necrosis, as illustrated in Figure 5A. All groups showed a significant reduction in overall liver scores compared to the induction group and considerably higher liver scores than the control group. Isofraxidin alone resulted in markedly higher liver scores compared to the P-MP group. However, the combination of isofraxidin and MP showed negligible differences compared to the P-MP group, as shown in Figure 5C.

## 4. Discussion

In the current investigation, isofraxidin decreases IL-6, TNF-α, IL-1β, IL-8, and INF-γ serum levels compared to the induction group. Also, isofraxidin lowers MDA and elevates GSH levels compared to the induction group, indicating that isofraxidin mitigates the onset and advancement of CRS caused by LPS in mice. Isofraxidin has been shown to prevent inflammatory disorders and associated organ injury by inhibiting the expression of pro-inflammatory cytokines in LPS-induced inflammation in mice [28]. Isofraxidin has anti-inflammatory effects both in vitro and in vivo through MAPK-dependent mechanisms. Findings from preclinical studies illustrated that the IP administration of isofraxidin before LPS administration prevents tissue injury and the development of hyper-inflammation by reducing the expression of NF-κB, IL-1β, IL-6, and TNF-α in the liver [28]. Hence, isofraxidin could be effective against LPS-induced inflammatory disorders. Niu et al. confirmed that isofraxidin attenuates the activation of peritoneal macrophages and the release of the TNF-α via MAPK-dependent mechanism [18].

Interestingly, isofraxidin ameliorates LPS-induced acute lung injury in mice by suppressing the expression of pro-inflammatory cytokines. It reduces IL-6 and TNF-α in the bronchial alveolar fluid and serum of mice treated with LPS. In addition, isofraxidin inhibits neutrophil myeloperoxidase and macrophage activation in lung tissues [42]. Furthermore, isofraxidin mitigates myocardial infarction by inhibiting the expression of NLRP3 inflammasome, which induces pro-inflammatory cytokine expression [43]. Moreover, isofraxidin has a potent antioxidant effect by reducing ROS generation and upregulating the antioxidant enzymes. Isofraxidin prevents dextran-induced ulcerative colitis by decreasing the development of oxidative stress through the modulation of nuclear erythroid-related factor 2 (Nrf2) and the production of oxidative species [44].

Similarly, isofraxidin is regarded as a potent scavenger of ROS and suppresses radiation-induced apoptosis through modulation of the ROS/mitochondrial pathway [45]. Another study showed that curcumin analogs exhibited antioxidant activity showed by enhanced catalase, SOD, and GSH, as well as reduced MDA levels in the scopolamine-induced stress mice model [46]. Another study examining mono-carbonyl curcumin analogs showed a significant reduction in MDA level and enhanced catalase, SOD, and GSH activities [47]. Therefore, because of its anti-inflammatory and antioxidant effects, isofraxidin ameliorates the propagation of CRS.

It has been shown that isofraxidin can prevent the development of CRS by inhibiting the expression of IL-6, the main pro-inflammatory cytokine involved in the development and progression of CRS [48]. Singh et al. suggest that isofraxidin could be effective against CRS in COVID-19 by suppressing SARS-CoV-2-induced hyper-inflammation, hypercytokinemia, and the development of CRS [49]. A preclinical study demonstrated that isofraxidin alleviates influenza-induced CRS in mice [50]. Mounting evidence from many studies has revealed that isofraxidin prevents influenza and SARS-CoV-2-induced acute lung injury and associated inflammatory and thrombotic disorders [17,51]. Likewise, isofraxidin protects the liver from high-fat-diet-induced hepatic inflammation in mice [52]. Additionally, isofraxidin protects different organ tissues, including the liver, from the effects of hyper-inflammation in experimental mice [53]. These findings indicate that isofraxidin has a potent effect against the development of CRS by regulating the immuno-inflammatory response and inhibiting oxidative stress. Hence, isofraxidin can protect organ tissues from the effects of CRS induced by LPS and viral infections.

In combination with methylprednisolone, isofraxidin decreased serum levels of IL-6, TNF-α, IL-1β, IL-8, and INF-γ in LPS-treated mice relative to the groups receiving either methylprednisolone or isofraxidin alone. Combining isofraxidin and methylprednisolone decreased MDA levels and elevated GSH levels compared to the monotherapy groups of methylprednisolone and isofraxidin.

The elevation of GSH levels post-treatment may possess several biological ramifications. GSH is an essential antioxidant that safeguards cells from oxidative stress and damage. Increased GSH levels signify that cells are more adept at neutralizing ROS and other free radicals; this may result in better cellular health and functionality, along with augmented detoxification mechanisms. Elevated GSH levels after treatment may indicate the efficacy of the intervention in enhancing the body’s antioxidant defenses; this is especially crucial in situations when oxidative stress is a prominent factor, such as in CRS. Moreover, elevated GSH levels facilitate the regeneration and repair of damaged tissues, as GSH participates in numerous cellular activities, including DNA synthesis and repair, protein synthesis, and enzyme activation, enhancing overall recovery and healing post-treatment [54,55].

Histopathologically, the combination group exhibited a protective benefit against experimental acute liver injury. However, this effect was not statistically significant compared to the methylprednisolone group. These data suggest that isofraxidin enhances methylprednisolone’s anti-inflammatory and antioxidant effects by diminishing pro-inflammatory cytokine expression and suppressing oxidative stress.

In the current investigation, isofraxidin decreased serum levels of IL-6, TNF-α, IL-1β, IL-8, and INF-γ in LPS-treated mice relative to the induction group. Furthermore, isofraxidin diminished MDA levels and enhanced GSH levels compared to the induction group. Isofraxidin can mitigate the onset and progression of CRS induced by LPS in mice, and it can reduce the development of acute liver injury, but not acute lung injury, in LPS-induced CRS in mice. Numerous studies have demonstrated the therapeutic effectiveness of isofraxidin in treating inflammatory diseases. Isofraxidin has been reported to decrease the inflammation generated by a high-fat diet in mice via modulating hepatic lipid metabolism, inflammatory responses, and the production of ROS.

Moreover, isofraxidin prevents the further exacerbation of inflammation caused by downregulating the NF-κB expression and releasing pro-inflammatory cytokines [52]. Niu et al. found that isofraxidin reduces the detrimental effect of LPS by inhibiting the MAPK signaling pathway, which activates inflammation and oxidative stress [18]. Of interest is that the administration of isofraxidin can reduce the propagation of inflammatory reactions in influenza virus infection by inhibiting platelet aggregation, which triggers the release of pro-inflammatory cytokines and MMP-9. Furthermore, isofraxidin decreases the progression of CRS by inhibiting the expression of pro-inflammatory cytokines and the progression of oxidative stress [49]. Therefore, treatment with isofraxidin may be effective against the development of acute lung and liver injury. Many phytochemicals, such as isofraxidin, have been revealed to reduce coronavirus-induced acute lung injury by inhibiting exaggerated oxidative and inflammatory stress [17]. The administration of isofraxidin consistently attenuates LPS-induced acute lung injury by decreasing the generation of pro-inflammatory cytokines and prostaglandin. Isofraxidin reduces prostaglandin secretion in bronchial alveolar fluid in mice with experimental acute lung injury [42]. Correspondingly, isofraxidin mitigates experimental acute liver injury by modulating oxidative stress, inflammatory reactions, and the expression of inflammatory signaling pathways. On the other hand, isofraxidin combined with methylprednisolone was more effective in reducing TNF-α, IL-8, INF-γ, and MDA as well as increasing GSH compared to the methylprednisolone-treated group. Therefore, the combined effect of isofraxidin and methylprednisolone seems more effective than that of isofraxidin and methylprednisolone alone.

Co-administering methylprednisolone with an additional medication to mitigate inflammation can augment the overall efficacy of treatment. Methylprednisolone is a corticosteroid that suppresses inflammation and regulates the immunological response. When utilized alongside other anti-inflammatory medicines, the treatment can tackle several mechanisms implicated in the inflammatory process, resulting in a more thorough and effective suppression of inflammation [24,27,30,56].

The present study’s findings showed that isofraxidin combined with methylprednisolone has potent anti-inflammatory and antioxidant effects in the induced CRS model; this is probably because isofraxidin potentiates the anti-inflammatory effect of methylprednisolone. Yarnell et al. reported that, combined with methylprednisolone, isofraxidin was more effective in mitigating the inflammatory reactions and biomarkers of oxidative stress than methylprednisolone in patients with rheumatoid arthritis [57].

There are limitations in the present study; while our results showed powerful antioxidant and anti-inflammatory effects, its exact molecular mechanisms need to be elucidated. The second limitation is that the current study covers the short-term outcomes; it did not cover subacute or chronic inflammatory responses. Finally, this study examined liver and lung tissue; it did not examine other tissues like the brain and kidney, which is necessary to obtain the full picture of the isofraxidin effect.

## 5. Conclusions

Isofraxidin exhibited preventive and therapeutic properties against lipopolysaccharide-induced cytokine storms in mice via anti-inflammatory and antioxidant pathways, and its combination with methylprednisolone demonstrated synergistic outcomes.

These findings suggest that isofraxidin has significant potential as a therapeutic agent for mitigating inflammation and oxidative stress in cytokine-release conditions like acute lung injury and systemic infection.

## Figures and Tables

**Figure 1 biomedicines-13-00653-f001:**
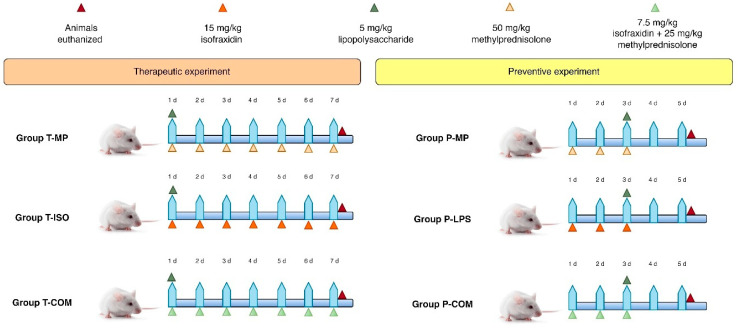
Experimental design.

**Figure 2 biomedicines-13-00653-f002:**
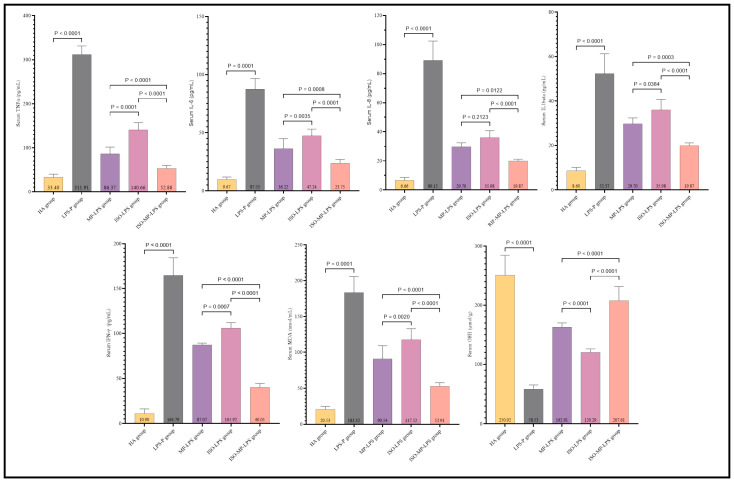
Isofraxidin’s protective function and synergistic effect with and without methylprednisolone on CRS inflammatory and oxidative stress markers.

**Figure 3 biomedicines-13-00653-f003:**
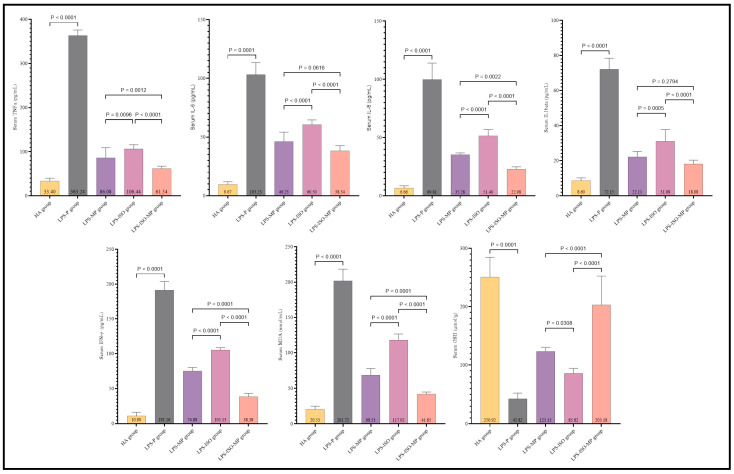
Isofraxidin’s therapeutic function and synergistic effect with and without methylprednisolone on CRS inflammatory and oxidative stress markers.

**Figure 4 biomedicines-13-00653-f004:**
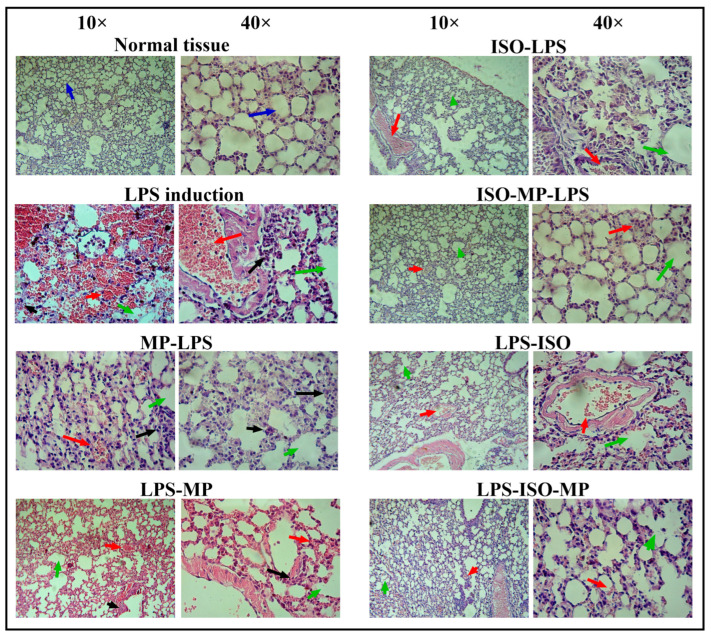
Histopathology of lung tissue showing impacts of different therapies in mice. H&E stain, 10× and 40× power. Blue arrow: normal lung architecture including alveoli; black arrow: severe acute inflammation; red arrow: vascular congestion and capillary destruction; green arrow: thick alveolar walls and narrow air space with hyaline membrane.

**Figure 5 biomedicines-13-00653-f005:**
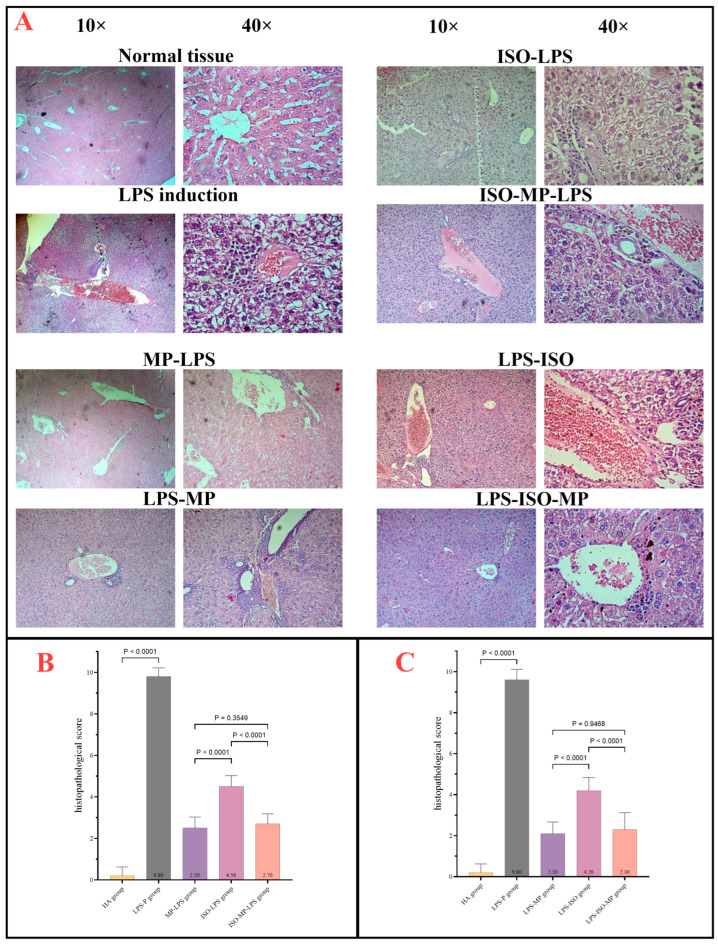
(**A**) Histopathology of liver tissue showing impacts of different therapies in mice. H&E stain, 10× and 40× power. (**B**) Liver damage score for preventive experiment. (**C**) Liver damage score for therapeutic experiment.

## Data Availability

The data presented in this study are available on request from the corresponding author (due to ongoing patency registration).

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
