# Peer review of "Isofraxidin Attenuates Lipopolysaccharide-Induced Cytokine Release in Mice Lung and Liver Tissues via Inhibiting Inflammation and Oxidative Stress"

_biomedicines, 2025, doi:10.3390/biomedicines13030653_

Round 1
Reviewer 1 Report
Comments and Suggestions for Authors
The manuscript “Isofraxidin potential as a therapeutic and preventive agent in LPS-induced cytokine releasing syndrome in mice through anti-inflammatory and antioxidant mechanism: an in vivo study” was thoroughly evaluated. The manuscript reported the antioxidant and anti-inflammatory potential of Isofraxidin in LPS-induced cytokine releasing syndrome in mice. The outline of the study is quite well and is well designed and written in scientific way. The author should address all the concerns before publishing.
1. Title of the manuscript is not attractive and appealing for the readers. So, I suggest to modify the title
2. Define all abbreviations at their first mention in the text and add a list of abbreviations for easy reference.
3. Line 61; remove the words Background/Objectives from the heading.
4. In the background of abstract the author should also mention about the inflammation for which the study was conducted.
5. The methods in the abstract were presented somewhat in details it should be revising and be presented briefly.
6. Results section of abstract lack numerical values from results of the study, the author should incorporate some values from key findings from results.
7. Introduction; the author should add few sentences about inflammation in general rather to specifically go for cytokine syndrome in the 1st para of introduction.
8. Line 61; the word “infection” is presented twice, it should be corrected.
9. Line 91; the abbreviation “SIRS” looks appear for first time; it should be described briefly to clarify the statement.
10. Figure 2; the resolution should be enhanced for better readability.
11. Figure 3; legends and values are not clear the author should enhance the pixel.
12. Figure 4; it should be better if the author mark the tissues with arrows and specify them in each illustration.
13. Figure 5 B and C; the values and legends are not clearly visible for readers, resolution should be enhanced.
14. The author should remove the subsections from discussion section.
15. In discussion section; after reference 37, the description of ROS should be expanded further in the light of updated references; for guidance, https://doi.org/ 10.3390/biomedicines10102597, https://doi.org/10.3390/ biomedicines10102385.
16. The author should expand conclusion section mentioning the future perspective of this study how it can benefit future research.
17. What breakthroughs do the authors think their research has made compared with the past?
18. The author should mention the limitation of the study.
19. Proofread the manuscript for grammatical errors, awkward phrasing, and typos errors.
20. Moreover, the similarity report suggests a higher percentage hence require lowering the similarity index.
21. References should be according to the journal format. Please correct reference number 28 as per format.
22. In my opinion, this article shows valuable information and should be revised accordingly for the aforesaid points before publishing.
Comments on the Quality of English LanguageProofread of the manuscript for grammatical errors, awkward phrasing, and typos errors before acceptance.
Author Response
Reviewer 1
Comment 1: Title of the manuscript is not attractive and appealing for the readers. So, I suggest to modify the title.
Answer
Thank you for pointing this out. We agree with the comment. Therefore, we have change the title to the following: “Isofraxidin attenuates lipopolysaccharide-induced cytokine releasing in mice lung and liver tissues via inhibiting inflammation and oxidative stress”.
Page: 1, lines: 1 to 4.
Comment 2: Define all abbreviations at their first mention in the text and add a list of abbreviations for easy reference
Answer
Thank you for pointing this out. We agree with the comment. Therefore, we have added a list of abbreviations and double-checked to see if we missed any.
The list of abbreviations:
“lipopolysaccharide (LPS), interleukin (IL), Malondialdehyde (MDA), Cytokine release syndrome (CRS), tumor necrosis factor (TNF), interferon (IFN), Nuclear factor kappa-light-chain-enhancer of activated B cells (NF-κB), cyclooxygenase 2 (COX-2), mitogen-activated protein kinase (MAPK), inducible nitric oxide synthase (iNOS), matrix metalloproteinases (MMPs), disinters metalloproteinase and thrombospondin motif 4,5 (ADAMTS-4,5), toll-like receptor 4 (TLR4), myeloid differentiation protein 2 (MD2), activating protein 1 (AP1), extracellular signal-regulated protein kinases 1 and 2 (ERK1/2), reactive oxygen species (ROS), lipopolysaccharide (LPS), enzyme-linked immunosorbent assay (ELISA), hematoxylin and eosin (H&E), methylprednisolone (MP), intraperitoneal (IP), glutathione (GSH)”
Pages 11-12, lines 366 - 377
Comments 3: Line 61; remove the words Background/Objectives from the heading
Answer
Thank you for pointing this out. According to the journal instructions, this head should be named Background/Objectives. We removed the objective and kept the background.
Page 1, line 16.
Comment 4: In the background of abstract the author should also mention about the inflammation for which the study was conducted.
Answer
Thank you for pointing this out. We mentioned inflammation in background, which can be found in page 1, line 17.
Comments 5: The methods in the abstract were presented somewhat in details it should be revising and be presented briefly.
Answer
Thank you for pointing this out. We agree with the reviewer, and we reduced the details of the methods while keeping the scientific content.
Comments 6: Results section of abstract lack numerical values from results of the study, the author should incorporate some values from key findings from results.
Answer
Thank you for pointing this out. We agree with the reviewer; however, due to the limited number of words in the abstract (250 words, and we currently have 230), adding numerical value will adversely affect the other content of the abstract.
Comments 7: Introduction; the author should add few sentences about inflammation in general rather to specifically go for cytokine syndrome in the 1st para of introduction
Answer
Thank you for pointing this out. We agree with the reviewer and add the required section.
Acute lung damage and acute respiratory distress syndrome are recognized as a category of pulmonary disease characterized by widespread inflammatory cell infiltration, hemorrhage, and pulmonary edema [1, 2]. Factors that might directly or indirectly generate these lung disorders include inhalation, pneumonia, sepsis, trauma, pancreatitis, and transfusion, among others [3]. Numerous studies indicate that inflammation is pivotal to the etiology of acute lung injury. A substantial influx of immune cells congregates at in-jury sites, triggering a cascade of inflammatory signaling pathways and releasing several pro-inflammatory cytokines. Consequently, inflammation compromises the barrier function of the alveolar epithelium and vascular endothelium, thereby enhancing the permeability of the alveolar-capillary membrane. This results in the accumulation of protein-rich edema fluid in the lung interstitium, ultimately causing pulmonary edema and tissue damage [4, 5].
Comments 8: Line 61; the word “infection” is presented twice, it should be corrected.
Answer
Thank you for pointing this out. We agree with the reviewer and corrected this typo error.
Comments 9: Line 91; the abbreviation “SIRS” looks appear for first time; it should be described briefly to clarify the statement.
Answer
Thank you for pointing this out. We removed this abbreviation.
Comment 10: Figure 2; the resolution should be enhanced for better readability.
Answer
Thank you for pointing this out. We agree with the reviewer and will submit each Figure with a higher resolution.
Comment 11: Figure 3; legends and values are not clear the author should enhance the pixel.
Answer
Thank you for pointing this out. We agree with the reviewer and will submit each Figure with a higher resolution.
Comment 12: Figure 4; it should be better if the author mark the tissues with arrows and specify them in each illustration.
Answer
Thank you for pointing this out. We agreed with the reviewer and added the arrow inside each figure.
Comment 13: Figure 5 B and C; the values and legends are not clearly visible for readers, resolution should be enhanced.
Answer
Thank you for pointing this out. We agree with the reviewer and will submit each Figure with a higher resolution.
Comment 14: The author should remove the subsections from discussion section.
Answer
Thank you for pointing this out. We agree with the reviewer and remove the subsections.
Comment 15: In discussion section; after reference 37, the description of ROS should be expanded further in the light of updated references; for guidance, https://doi.org/ 10.3390/biomedicines10102597, https://doi.org/10.3390/ biomedicines10102385.
Answer
Thank you for pointing this out. We agree with the reviewer and expanded the discussion based on these two references.
Comment 16: The author should expand conclusion section mentioning the future perspective of this study how it can benefit future research.
Answer
Thank you for pointing this out. We agree with the reviewer and expand the conclusions.
Comment 17: What breakthroughs do the authors think their research has made compared with the past?
Answer
Thank you for pointing this out. No study in literature examined the effect of isofraxidin on CRS. Due to its multiple mechanisms of action, including anti-inflammatory and antioxidant activity, this action could benefit CRS, which develops due to an exaggerated immune response and is associated with hyperinflammation and hypercytokinemia. Targeting inflammatory signaling pathways and proinflammatory cytokines by pleiotropic anti-inflammatory action of isofraxidin.
Comment 18: The author should mention the limitation of the study?
Answer
Thank you for pointing this out. We added study limitations at the end of the discussion.
Comment 19: Proofread the manuscript for grammatical errors, awkward phrasing, and typos errors.
Answer
Thank you for pointing this out. We improved the language and corrected grammatical errors, awkward phrasing, and typos errors.
Comment 20: Moreover, the similarity report suggests a higher percentage hence require lowering the similarity index.
Answer
Thank you for pointing this out. We reduced the similarity.
Comment 21: References should be according to the journal format. Please correct reference number 28 as per format
Answer
Thank you for pointing this out. We used the EndNote program to write all the references and corrected reference 28.
Ramírez K, Quesada-Yamasaki D, Jaime F-TC. A Protocol to Perform Systemic Lipopolysacharide (LPS) Challenge in Rats. Odovtos. 2019;21(1):53-66.
Comment 22: In my opinion, this article shows valuable information and should be revised accordingly for the aforesaid points before publishing.
Answer
Thank you for pointing this out. All the requested revisions were done following reviewer recommendations.
Reviewer 2 Report
Comments and Suggestions for Authors
The study is well-organized, and the manuscript is written comprehensively.
The authors should address the biological indication of the increase of GSH after treatment in Discussion. Also, the purpose of the combination of methylprednisolone in this study is for those readers who are not familiar with the research area.
Author Response
Comment 1: The authors should address the biological indication of the increase of GSH after treatment in Discussion.
Answer
Thank you for pointing this out. We added the biological indication of increasing GSH after treatment in the discussion, as follows:
The elevation of GSH levels post-treatment may possess several biological ramifications. GSH is an essential antioxidant that safeguards cells from oxidative stress and damage. Increased GSH levels signify that cells are more adept at neutralizing ROS and other free radicals; this may result in better cellular health and functionality, along with augmented detoxification mechanisms. Elevating GSH levels after treatment may indicate the efficacy of the intervention in enhancing the body’s antioxidant defenses; this is especially crucial in situations when oxidative stress is a prominent factor, such as in CRS. Moreover, elevated GSH levels facilitate the regeneration and repair of damaged tissues, as GSH participates in numerous cellular activities, including DNA synthesis and repair, protein synthesis, and enzyme activation, enhancing overall recovery and healing post-treatment [52, 53].
Comment 2: Also, the purpose of the combination of methylprednisolone in this study is for those readers who are not familiar with the research area.
Thank you for pointing this out. We added the following section in the discussion, as follows:
Co-administering methylprednisolone with an additional medication to mitigate in-flammation can augment the overall efficacy of treatment. Methylprednisolone is a cor-ticosteroid that suppresses inflammation and regulates the immunological response. When utilized alongside other anti-inflammatory medicines, the treatment can tackle sev-eral mechanisms implicated in the inflammatory process, resulting in a more thorough and effective suppression of inflammation [27, 30, 54, 55].
Reviewer 3 Report
Comments and Suggestions for Authors
The paper of Marwa Salih Al-Naimi «Isofraxidin potential as a therapeutic and preventive agent in LPS-induced cytokine releasing syndrome in mice through anti-inflammatory and antioxidant mechanism: an in vivo study» is devoted to the biological action of isofraxidin, which is a herbal extract with a broad spectrum of activity.
The study aimed to examine the therapeutic effects of isofraxidin with or without methylprednisolone to ameliorate lipopolysaccharide (LPS) induced cytokine-releasing syndrome. The authors of the article comprehensively investigated the therapeutic effect of isofraxidin and, based on the results of the work, a conclusion was made about the prospects of this compound against cytokine storms caused by lipopolysaccharide in mice through anti-inflammatory and antioxidant pathways, and its combination with methylprednisolone demonstrates synergistic results.
The analyzed literary sources show good knowledge of the studied problem and possible ways of therapeutic improvements. The Materials and Methods provide a clear presentation of the research methods and experimental data. The graphic material is presented in the form of high-quality drawings that fully reflect the content of the research.
While reading the manuscript, a number of questions arose:
1) How was the purity of isofraxidin isolated from the extract confirmed (that it is not a mixture of different organic compounds)?
Typos:
«…of 7.5 mg/kg…» missing space; 2 ml (should be mL),
So, I recommend an article for publication in Biomedicines after revision.
Author Response
The paper of Marwa Salih Al-Naimi «Isofraxidin potential as a therapeutic and preventive agent in LPS-induced cytokine releasing syndrome in mice through anti-inflammatory and antioxidant mechanism: an in vivo study» is devoted to the biological action of isofraxidin, which is a herbal extract with a broad spectrum of activity.
The study aimed to examine the therapeutic effects of isofraxidin with or without methylprednisolone to ameliorate lipopolysaccharide (LPS) induced cytokine-releasing syndrome. The authors of the article comprehensively investigated the therapeutic effect of isofraxidin and, based on the results of the work, a conclusion was made about the prospects of this compound against cytokine storms caused by lipopolysaccharide in mice through anti-inflammatory and antioxidant pathways, and its combination with methylprednisolone demonstrates synergistic results.
The analyzed literary sources show good knowledge of the studied problem and possible ways of therapeutic improvements. The Materials and Methods provide a clear presentation of the research methods and experimental data. The graphic material is presented in the form of high-quality drawings that fully reflect the content of the research.
While reading the manuscript, a number of questions arose:
Comment 1: How was the purity of isofraxidin isolated from the extract confirmed (that it is not a mixture of different organic compounds)?
Answer
Thank you for pointing this out. Isofraxidin was purchased from Hangzhou hyper chem. Limited, China as power with 98.0% purity. We added this information to the methods section.
Comment 2: Typos: «…of 7.5 mg/kg…» missing space; 2 ml (should be mL),
Answer
Thank you for pointing this out. We corrected this error.